# Effect of a Long-Term Online Home-Based Supervised Exercise Program on Physical Fitness and Adherence in Breast Cancer Patients: A Randomized Clinical Trial

**DOI:** 10.3390/cancers16101912

**Published:** 2024-05-17

**Authors:** María Elena Garcia-Roca, Ignacio Catalá-Vilaplana, Carlos Hernando, Pablo Baliño, Pablo Salas-Medina, Pilar Suarez-Alcazar, Ana Folch-Ayora, Eladio Collado Boira

**Affiliations:** 1Faculty of Health Sciences, Pre-Department of Nursing, Jaume I University, 12071 Castellon, Spain; garciroc@uji.es (M.E.G.-R.); psalas@uji.es (P.S.-M.); malcazar@uji.es (P.S.-A.); afolch@uji.es (A.F.-A.); colladoe@uji.es (E.C.B.); 2Department of Physical Education and Sports Sciences, University of Valencia, 46010 Valencia, Spain; 3Sport Service, Department of Education and Specifics Didactics, Jaume I University, 12071 Castellon, Spain; hernando@uji.es; 4Faculty of Health Sciences, Pre-Department of Medicine, Jaume I University, 12071 Castellon, Spain; balino@uji.es

**Keywords:** breast neoplasm, treatment, physical function, exercise oncology, physical activity

## Abstract

**Simple Summary:**

One of the most prevalent malignancies across the world is female breast cancer, accounting for 25% of all diagnosed cancers. Physical exercise has been recognized as an important strategy for prevention and treatment during the cancer continuum. Home-based exercise programs can produce greater adherence rates than in-person interventions. However, the majority of home-based programs are carried out employing practical guides, brochures or electronic materials without supervision, which can increase the risk of injury and adverse effects. The aim of this study was to analyze the effect of a synchronous-supervised online home-based exercise program during 24 weeks on body composition, physical fitness and adherence compared to an exercise recommendation group without supervision with patients undergoing breast cancer treatment. We confirmed that supervised home-based exercise interventions can be an interesting strategy to improve physical fitness and adherence rates in breast cancer patients undergoing treatment.

**Abstract:**

The purpose of the present study was to analyze the effect of a synchronous-supervised online home-based exercise program (HBG) during 24 weeks on body composition, physical fitness and adherence compared to an exercise recommendation group (ERG) without supervision with patients undergoing breast cancer treatment. Fifty-nine female breast cancer patients (31 in HBG and 28 in the ERG) undergoing cancer treatments participated in the present randomized clinical trial. The exercise program consisted of a 60 min combined resistance and aerobic supervised exercise session (6–8 points on Borg Scale CR-10, moderate intensity), twice a week during 24 weeks. The exercise recommendation group only received general recommendations to comply with the current ACSM guidelines. Body composition and physical fitness were assessed at baseline, 12 weeks and 24 weeks of the program. Adherence to the intervention was measured according to the minutes of exercise completed per session during each week. A general linear model of two-way repeated measures showed significant improvements (*p* < 0.05) in physical fitness that were observed in the home-based exercise group at the baseline, 12-week and 24-week assessments compared to the exercise recommendation group. Adherence was also higher in the home-based exercise group. However, no changes (*p* > 0.05) in body composition between groups and moments were observed. In this sense, supervised home-based exercise interventions can be an interesting strategy to improve physical fitness and adherence rates in breast cancer patients undergoing treatment.

## 1. Introduction

One of the most prevalent malignancies across the world is female breast cancer, accounting for 25% of all diagnosed cancers [1]. Survival rates have been increasing in recent years due to early detection and treatment improvements [2]. However, the administered treatments provoke important side effects, such as fatigue, sarcopenia, osteoporosis, cardiac toxicity, joint pain and, in general, a decrease in quality of life [3,4].

Physical exercise has been recognized as an important strategy for prevention and treatment during the cancer continuum [5], with the development of international guidelines [6]. In addition to the association with reduced risk of different types of cancer [7], exercise has been demonstrated during and after cancer treatment to improve overall fitness, as well as to counterbalance side effects from pharmacological and surgical treatments [8,9]. Different studies have also been exploring the influence of physical exercise on the reduced risk of breast cancer recurrence [9,10], highlighting the importance of an active lifestyle for individuals living with and beyond cancer [8]. Despite this fact, a large percentage of the patients (93%) are not sufficiently active [11].

Most of the exercise programs have been conducted using supervised, 1:1 in-person interventions [12,13]. However, such programs can generate low training adherence rates due to different barriers related to economic status, aesthetic factors and treatment side effects [9,11,14]. To address this issue, an assessment of exercise interventions that have fewer barriers and require fewer resources for patients undergoing active treatment but still provide health benefits is needed [15].

Home-based exercise programs have been reported to have superior adherence [16], while group-based programs require fewer resources than 1:1 supervised in-person training and provide the beneficial effects of group dynamics [15]. These types of programs have been demonstrated to be an effective and safe strategy to improve fatigue, quality of life and functional capacity in cancer patients [17,18,19]. Nevertheless, training variables like intensity, volume or technique should be adapted and controlled for each patient [20], since the majority of home-based programs are carried out employing practical guides, brochures or electronic materials without supervision, which can increase the risk of injury and adverse effects [21,22,23]. In this sense, it is crucial to develop supervised exercise programs adapted to the personal characteristics of each cancer patient in order to improve physical fitness.

Therefore, the purpose of the present study was to analyze the effect of a synchronous-supervised online home-based exercise program during 24 weeks on body composition, physical fitness and adherence compared to an exercise recommendation group without supervision with patients undergoing breast cancer treatment. It was hypothesized that (a) benefits in body composition, physical fitness and adherence would be higher in the home-based exercise group in comparison with the exercise recommendation group (H_1_), and (b) body composition and physical fitness would improve after 24 weeks of home-based exercise or exercise recommendations compared to baseline (H_2_).

## 2. Materials and Methods

### 2.1. Study Design

This is a randomized clinical trial with two groups, the synchronous-supervised online home-based exercise group and the exercise recommendation group. Participants were randomized (1:1) by the oncologist at the time of diagnosis, being included in the home-based exercise group or the exercise recommendation group, alternatively. The intervention had a duration of 24 weeks, with a baseline assessment carried out in November 2021, with a 12-week assessment performed in February 2022 and a 24-week assessment completed in May 2022.

Participants were recruited from the Hospital Provincial de Castellón (Castellón, Spain). The study was registered at ClinicalTrials.gov with trial registration number NCT06275321. Participants agreed to participate in the study and provided written informed consent. The study procedures complied with the Declaration of Helsinki and were approved by the Jaume I University ethics committee (CD/55/2019).

### 2.2. Participants

Sixty-one breast cancer patients (thirty-one participants in the home-based exercise group and thirty participants in the exercise recommendation group) were selected to participate in the study. Two abandoned the study due to personal reasons. Therefore, the final sample included fifty-nine female patients (thirty-one participants in the home-based exercise group and twenty-eight participants in the exercise recommendation group) who participated in the study. Eligibility criteria included being 18 years old or older, being diagnosed with breast cancer, undergoing cancer treatment (chemotherapy, hormone therapy, radiotherapy or immunotherapy) and having no medical contraindications for exercise practice (e.g., cardiovascular disease or neuromuscular disorders).

Consecutive non-probabilistic sampling with an exact estimation of the sample size through the Granmo calculator was performed, using data provided by the breast oncology surgery services, with an annual average of 220 women diagnosed with breast cancer and undergoing systemic oncological treatment. With the volume of patients, a confidence level of 95%, a precision of 5% and an expected proportion of losses of 15% (women who do not complete the study), were obtained for a sample of *N* = 59 patients. This is considered optimal as it is consistent with previous studies related to the same topic [24,25,26].

### 2.3. Procedure

The synchronous-supervised home-based group participated in an online exercise program in streaming supervised by their oncology team for 24 weeks. Participants were asked to engage in a 60 min combined resistance and aerobic exercise session twice a week for 24 weeks. All the sessions were developed through Google Meet (Google, Sunnyvale, CA, USA) video-callings. The sessions were controlled, led and supervised by an exercise specialist for cancer patients, who encouraged and gave feedback to the participants, while they could watch the performance, interact or ask questions.

The sessions consisted of a 10 min warm-up, with joint mobility and balance exercises. Then, the main part was completed for 40 min in order to improve upper and lower body strength and cardiorespiratory fitness, focusing on all major muscle groups and using body weight, exercise mats, resistance bands and/or free weights. This part included a combined circuit of 8–12 functional exercises (e.g., squats, front and side lunges, sit-ups, calf raises, glute bridge, core, biceps curl, shoulder press, punches, jumping jacks and static walking/jogging). The circuit included 2 series of 10–12 repetitions for the functional strength exercises and 30 s for the aerobic exercises. The volume increased progressively by modifying the number of repetitions and sets and the complexity of the exercises. A minimum rest of 30 s between exercises and 90 s between sets was established. The last 10 min (cool down) included stretching exercises for the major muscle groups, breathing and relaxation techniques. At the end of each session, a fatigue scale (Borg Scale CR-10) [27] was applied, and the intensity of the subsequent sessions was programmed based on the percentage of the rating of perceived exertion reached. The intensity was adapted to reach a rating of perceived exertion values between 6 and 8 points on Borg Scale CR-10 (moderate intensity).

The exercise recommendation group only received general recommendations to comply with the current ACSM (American College of Sports Medicine) guidelines [6]. These guidelines were explained individually by the exercise specialist during the baseline assessments to promote awareness of the benefits of physical exercise in cancer patients. Patients were instructed to continue their usual activities and received explanatory videos in order to help them with their workouts, but it was not supervised. Physical activity levels were monitored through telephone calls, text messaging and e-mail every week to follow up on the patients’ progress and health status and motivate them to continue exercising. After completion of the exercise program, participants in the exercise recommendation group were encouraged to adopt a more physically active lifestyle and were given the same guidance and physical exercise program as the intervention group.

### 2.4. Outcome Assessments

Outcomes included sociodemographic variables: age (years) and marital status (single, married, divorced, widow and others); clinical variables: tumor type, laterality and tumor stage; and treatment received: chemotherapy, radiotherapy and hormone therapy.

Anthropometric and body composition variables, body mass index, body fat percentage and muscle mass percentage, were measured at baseline, 12 weeks and 24 weeks of the exercise program. Body composition was determined through bioelectrical impedance analysis (BIA) (Tanita BC-780MA, Tanita Corp., Tokyo, Japan).

Also, physical fitness variables, heart rate, rating of perceived exertion, blood pressure, oxygen saturation, cardiorespiratory fitness, strength and flexibility, were measured at baseline, 12 weeks and 24 weeks of the exercise program. Cardiorespiratory fitness was assessed with the 6 min walking test, a widely used and validated measure in people with cancer [28,29]. This test was held on a 50 m rectangular circuit, with the aim of covering the maximum distance possible without running [30,31,32]. Upper-limb strength was determined using a handgrip dynamometer (Grip Strength Dynamometer, Takkei TKK 5101, Tokio, Japón) [33,34], while lower-limb strength was measured through a squat–jump test (SJ) and countermovement jump test (CMJ) [26,35] using a contact platform (Chronojump-Boscosystem, Barcelona, Spain) and the chair–stand test [23,36]. From a standing position, participants were asked to repeatedly sit down and stand up as fast as possible for 30 s. The number of stands was recorded [37]. Flexibility was registered with the sit-and-reach test [38]. CMJ, SJ, handgrip (both hands) and sit-and-reach tests were repeated three times, and the best attempt of each test was selected for further analysis. Two minutes between tests were allowed in order to avoid the effect of fatigue. Adherence was measured according to the minutes of exercise completed per session during each week [39].

### 2.5. Statistical Analysis

Statistical analysis was carried out using the SPSS.29 statistics software package (SPSS Inc., Chicago, IL, USA). The data normality and homoscedasticity were verified using the Kolmogorov–Smirnov test. Then, a general linear model of a two-way repeated-measures design was performed. Groups (home-based and exercise recommendation) and moment (baseline, 12 weeks, 24 weeks) were considered, as within-subject factors. Post hoc comparisons were performed by applying the Bonferroni correction to identify the location of specific differences. The mean and standard deviation are presented for continuous variables. The level of significance was set at *p* < 0.05. For significant pair differences, effect size (ES) was assessed using Cohen’s *d* (0.2, small; 0.5, moderate; 0.8, large) [40].

## 3. Results

### 3.1. Participants Characteristics

Sociodemographic characteristics (age and marital status) and clinical variables (tumor type, laterality, tumor stage and treatment) are presented in Table 1.

### 3.2. Anthropometry and Body Composition

Anthropometric and body composition variables are presented in Table 2. No statistically significant differences (*p* > 0.05) were found in body mass index, body fat percentage or muscle mass percentage between groups (home-based exercise group vs. exercise recommendation group) or between moments (baseline, 12 weeks, 24 weeks).

### 3.3. Physical Fitness

Physical fitness results for the home-based exercise group and exercise recommendation group at baseline, 12 weeks and 24 weeks are presented in Table 3. Significant differences (*p* < 0.05) were found between groups in right and left handgrip and 6 min walking tests, observing better outcomes in the home-based exercise group at the baseline, 12-week and 24-week assessments.

Significant (*p* < 0.05) improvements were found in the home-based exercise group for the chair–stand test, sit-and-reach test, squat–jump test, countermovement jump test and 6 min walking test during the exercise program, while the exercise recommendation group just improved the chair–stand test and decreased significantly (*p* < 0.05) in the sit-and-reach test and countermovement jump test.

The rating of perceived exertion was significantly higher in the home-based exercise group at 12 weeks (*p* < 0.001, ES = 3.8) and 24 weeks (*p* < 0.001, ES = 3.8) (7.5 points and 6.1 points on Borg Scale CR-10, respectively) compared to the exercise recommendation group (2.2 points and 1.8 points on Borg Scale CR-10, respectively).

### 3.4. Safety and Adherence

No adverse events or health issues during the exercise intervention in the home-based group were noted. Adherence to the intervention averaged 111.1 min per week during the first 12 weeks and 114.0 min per week during the whole home-based exercise program (24 weeks), while adherence of the exercise recommendation group was 37.7 and 35.1 min per week (12 and 24 weeks, respectively).

## 4. Discussion

The main objective of the present study was to analyze the effect of a synchronous-supervised online home-based exercise program during 24 weeks on body composition, physical fitness and adherence compared to an exercise recommendation group without supervision in patients undergoing breast cancer treatment. Most of the exercise programs have been conducted using supervised, 1:1 in-person interventions, while the majority of home-based programs are carried out employing practical guides, brochures or electronic materials without supervision. Based on the results, we partially accept H_1_, since benefits in physical fitness and adherence were higher in the home-based exercise group in comparison with the exercise recommendation group, but no changes in body composition were observed. Also, H_2_ is partially accepted because physical fitness variables improved after 24 weeks of home-based exercise or exercise recommendations compared to baseline, but no differences were found in body composition variables.

Home-based exercise programs have been shown to be a valid strategy to improve body composition, strength, cardiorespiratory fitness and quality of life in cancer patients [18,41]. Previous studies found an improvement in body mass and body mass index after completing a 6-month home-based intervention in breast cancer survivors, but no differences in body fat percentage were observed [25]. The present study, in line with other investigations [23], did not find any changes between groups in body mass index, body fat percentage or muscle mass percentage after 24 weeks of supervised home-based exercise training or following exercise recommendations, which could be a positive outcome since a reduction in muscle mass has been associated with dependency, more functional limitations and lower cancer survival rates [42,43]. It should be highlighted that in the study by Lahart et al. [25], the intervention group followed a face-to-face consultation and support telephone call, which would be more similar to our exercise recommendation group rather than the home-based exercise group.

The supervised home-based exercise program improved physical fitness compared to the exercise recommendation group in breast cancer patients undergoing treatment. Significant improvements in the chair–stand test (12 weeks: 7 reps, 24 weeks: 10 reps), sit-and-reach test (12 weeks: 8.2 cm, 24 weeks: 19.6 cm), squat–jump test (12 weeks: 4.0 cm, 24 weeks: 4.7 cm), countermovement jump test (12 weeks: 4.2 cm, 24 weeks: 4.9 cm) and 6 min walking test (12 weeks: 160 m, 24 weeks: 265 m) in the online home-based group after 24 weeks of exercise training were found in comparison with the exercise recommendation group. These results are in overall agreement with those of Jones et al. [44], who also found a higher improvement in physical fitness in the supervised exercise group compared to the control group after 12 weeks.

One of the main side effects in breast cancer patients is related to skeletal muscle decline [45]. Previous studies have shown a pronounced muscle strength decline in breast cancer survivors [43,46,47]. In this study, both groups increased the number of repetitions after 24 weeks (home-based group: 15 reps, exercise recommendation group: 3 reps), showing better lower-limb muscle strength and balance at the 24-week assessment [37]. The home-based exercise group also showed an improvement in both squat–jump and countermovement jump tests (3.1 cm, and 4.0 cm, respectively) and in flexibility (9.7 cm) after an exercise intervention of 24 weeks. In line with these results, DeNysschen et al. [48], found improvements in handgrip strength, chair–stand and arm curl tests after an 8-week home-based exercise program in female breast cancer survivors. Sagarra-Romero et al. [23] also found improvements in handgrip strength (right hand) test, chair–stand test and cardiorespiratory fitness recorded by the Rockport test in breast cancer survivors after 16 weeks of supervised home-based exercise. Nevertheless, few studies have analyzed the effect of a supervised home-based exercise intervention during cancer treatment. However, the decline in physical fitness in the exercise recommendation group was evidenced by a decrease in flexibility and countermovement jump test after 24 weeks due to treatment side effects.

In terms of upper limbs’ strength, no differences in handgrip strength were observed between both groups after 24 weeks of supervised home-based exercise training or following exercise recommendations. According to Murri et al. [49], this could be due to the limited overloads used during the 24-week program (1–3 kg), especially because trainings were administered by a video call due to the COVID-19 pandemic, and it was necessary to ensure that the exercises were performed safely. Moreover, this intervention was designed to recover the function of the operated limb, not to improve the upper limbs’ strength [49].

The 6 min walking test, as an indicator of general health in breast cancer patients [28], showed an improvement in functional capacity in the home-based exercise group, since this group showed an increase of 178 m after 24 weeks of supervised exercise training, while the exercise recommendation group only improved by 7 m. Other authors also reported an increase in cardiorespiratory fitness assessed by the 6 min walking test after a 16-week supervised exercise program compared to a usual care group [49].

Finally, in-person interventions are associated with low training adherence rates [11], while home-based exercise programs have been shown to have superior adherence [16]. The adherence of the home-based exercise group in this study was very high in comparison with the exercise recommendation group (114 min vs. 35 min per week during 24 weeks, respectively), which demonstrates that supervised home-based exercise interventions can be an interesting strategy to improve physical fitness and adherence rates in breast cancer patients undergoing treatment [50].

The main limitation of the present investigation is related to the COVID-19 pandemic. The exercise program followed in this study was designed to be carried out in person. However, the program had to be adapted to an online version when the COVID-19 pandemic started. This limitation can also be one of the strengths of the study since the adherence of participants to the online supervised home-based exercise intervention was higher than other in-person programs and it offers greater flexibility and accessibility for those patients who have difficulties and barriers to attending in-person programs due to different side effects. Moreover, most of the studies have been carried out during 8 or 12 weeks, but this investigation completed a 24-week exercise program since we believed less than 24 weeks would not be sufficient to appreciate meaningful changes in physical fitness. It should be highlighted that the mean age of breast cancer patients who participated in this study does not represent the mean age of breast cancer patients in general, since all participants in this study were derived from oncologists and they just recommended the investigation to those who they believed could complete the exercise program.

## 5. Conclusions

A supervised home-based exercise program improved physical fitness compared to an exercise recommendation group in patients undergoing breast cancer treatment. Specifically, the home-based program improved the chair–stand test, flexibility, squat–jump test, countermovement jump test and 6 min walking test after 24 weeks of exercise training. The exercise recommendation group just showed improvement with the chair–stand test, with significantly decreased flexibility and countermovement jump test due to treatment side effects. Adherence rates were also higher in the home-based exercise group. However, no statistically significant differences were found in body composition between the home-based exercise group and the exercise recommendation group after 24 weeks of exercise training or exercise recommendations, respectively. Therefore, supervised home-based exercise interventions can be an interesting strategy to improve physical fitness and adherence rates in breast cancer patients undergoing treatment.

## Figures and Tables

**Table 1 cancers-16-01912-t001:** Sociodemographic characteristics and clinical variables in the home-based exercise group and the exercise recommendation group at baseline.

	Home-Based Exercise Group *N* (%)	Exercise Recommendation Group *N* (%)
Age (years, mean ± SD)	49.0 ± 8.9	50.1 ± 7.9
Marital Status		
Married or in a relationship	24 (77.4)	17 (60.7)
Separated or divorced	4 (12.9)	5 (17.9)
Single	2 (6.5)	6 (21.4)
Widowed	1 (3.2)	0 (0.0)
Breast cancer subtype		
Luminal A	12 (38.7)	10 (35.7)
Luminal B (her2 +)	4 (12.9)	5 (17.8)
Luminal B (her2 −)	11 (35.4)	10 (35.7)
Enriched-her2	2 (6.5)	2 (7.2)
Basal-like	2 (6.5)	1 (3.6)
Laterality		
Right breast	10 (32.3)	10 (35.7)
Left breast	17 (54.8)	17 (60.79
Bilateral	4 (12.9)	1 (3.6)
Tumor stage		
I	9 (29.0)	13 (46.4)
II	20 (64.5)	12 (42.9)
III	1 (3.2)	1 (3.6)
IV	1 (3.2)	2 (7.1)
Treatment during the study		
Chemotherapy	16 (51.6)	17 (60.7)
Radiotherapy	4 (13.0)	2 (7.1)
Hormone therapy	11 (35.5)	9 (32.2)

**Table 2 cancers-16-01912-t002:** Evolution of anthropometric and body composition variables in the home-based exercise group and the exercise recommendation group.

Home-Based Exercise Group	Baseline	*p* Value/ES	12 Weeks	*p* Value/ES	24 Weeks	*p* Value/ES
**Body Mass Index (Weight (kg)/Height (m^2^))**	25.7 ± 6.7	1.000	25.7 ± 6.4	0.250	22.5 ± 10.4	1.000
**Body Fat Percentage (%)**	34.7 ± 8.0	1.000	34.7 ± 7.8	1.000	33.2 ± 7.7	1.000
**Muscle Mass Percentage (%)**	62.2 ± 8.7	0.365	62.0 ± 7.7	1.000	63.5 ± 8.0	0.955
**Exercise recommendation group**						
**Body Mass Index (Weight (kg)/Height (m^2^))**	25.1 ± 4.2	1.000	24.8 ± 4.4	0.250	25.4 ± 4.6	1.000
**Body Fat Percentage (%)**	33.0 ± 7.1	1.000	33.1 ± 6.8	1.000	34.7 ± 7.1	1.000
**Muscle Mass Percentage (%)**	63.8 ± 6.7	0.365	60.8 ± 11.2	1.000	61.0 ± 8.0	0.955

Values are presented as mean ± standard deviation. ES: Effect Size.

**Table 3 cancers-16-01912-t003:** Evolution of physical fitness levels in the home-based exercise group and the exercise recommendation group.

Home-Based Exercise Group	Baseline	*p* Value/ES	12 Weeks	*p* Value/ES	24 Weeks	*p* Value/ES
**Right Handgrip (kg)**	25.2 ± 5.1 *	1.000	26.0 ± 4.2 *	0.804	26.8 ± 4.5 *	1.000
**Left Handgrip (kg)**	24.1 ± 5.8 *	1.000	25.2 ± 4.6 *	1.000	25.9 ± 4.3 *	1.000
**Chair–stand test (repetitions in 30″)**	18.0 ± 7.0 ^b^	<0.001/1.5	27.0 ± 5.0 * ^c^	<0.001/1.1	33.0 ± 6.0 * ^a^	<0.001/2.4
**Sit-and-reach test (cm)**	4.7 ± 8.0 ^b^	<0.001/1.0	11.3 ± 5.9 *	0.090	14.4 ± 4.6 * ^a^	<0.001/1.5
**Squat–jump test (cm)**	14.0 ± 5.9 ^b^	<0.001/0.4	16.1 ± 5.3 * ^c^	0.034/0.2	17.1 ± 5.1 * ^a^	<0.001/0.6
**Countermovement jump test (cm)**	13.3 ± 5.4 ^b^	<0.001/0.5	16.3 ± 6.2 *	0.052	17.3 ± 5.9 * ^a^	<0.001/0.7
**6 min walking test (m)**	686.2 ± 169.0 * ^b^	<0.001/0.6	789.3 ± 195.6 * ^c^	<0.001/0.4	863.7 ± 174.5 * ^a^	<0.001/1.1
**Exercise recommendation group**						
**Right Handgrip (kg)**	21.8 ± 4.8	1.000	20.9 ± 4.5	0.804	19.4 ± 5.4	1.000
**Left Handgrip (kg)**	19.9 ± 4.6	1.000	18.4 ± 5.3	1.000	17.8 ± 4.5	1.000
**Chair–stand test (repetitions in 30″)**	19.0 ± 5.0	0.752	20.0 ± 5.0 ^c^	0.001/0.5	23.0 ± 5.0 ^a^	<0.001/0.7
**Sit-and-reach test (cm)**	3.4 ± 10.3	1.000	3.1 ± 8.7 ^c^	<0.001/1.0	−5.2 ± 8.7 ^a^	<0.001/0.9
**Squat–jump test (cm)**	13.1 ± 4.3	0.067	12.1 ± 3.9	1.000	12.4 ± 3.8	0.458
**Countermovement jump test (cm)**	13.5 ± 4.8 ^b^	0.003/0.3	12.1 ± 4.4	1.000	12.2 ± 3.9 ^a^	0.040/0.3
**6 min walking test (m)**	600.9 ± 75.1	0.523	629.6 ± 111.0	1.000	607.7 ± 87.2	0.117

Values are presented as mean ± standard deviation. ES: Effect Size. * Significantly different (*p* < 0.05) compared to the exercise recommendation group. ^a^ Significantly different with Baseline; ^b^ Significantly different with 12 weeks; ^c^ Significantly different with 24 weeks.

## Data Availability

The data that support the findings of this study are available on request from the corresponding author. The data are not publicly available due to privacy or ethical restrictions.

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
