# Peer review of "Effect of a Long-Term Online Home-Based Supervised Exercise Program on Physical Fitness and Adherence in Breast Cancer Patients: A Randomized Clinical Trial"

_cancers, 2024, doi:10.3390/cancers16101912_

Round 1
Reviewer 1 Report
Comments and Suggestions for Authors
Dear authors,
Your manuscript is interesting but I need there are errors that cannot be corrected in the design.
ABSTRACT
A brief description of the analyses carried out is recommended.
INTRODUCTION
The description of barriers is not sufficiently detailed. In the case of home-based exercise, the same barriers may exist. What makes the home a better setting than a sports centre? In these centres there are professionals trained to carry out this supervision.
It is difficult to believe that exercise at home is more adherent than in sports centres. The literature provided is scarce and outdated. Justify this topic better.
MATERIALS AND METHODS
2.1. Study design:
All clinical trials are prospective. The term "prospective" should only be used in observational studies. Authors should correct this.
The authors have not said on what date the study was conducted.
2.3. Procedure:
The use of different items to perform the exercise involves a bias. For example, one patient performs biceps contraction with bags and another with bottles or milk cartons. The authors have not exposed the elements used by each individual and have not explained how to avoid this bias.
There was no face-to-face session where the exercises to be performed at home were explained. It is strange that all the patients knew how to do the exercises without making mistakes.
RESULTS
The authors have not included the dropout or "experimental death" rate of the study.
There are no statistically significant differences in many results. In addition, some tests cannot be monitored online. For example, how do you measure the 6 km walk test? or the vertical jump in centimeters? or the squat (well done) with jump in centimeters?
DISCUSSION
As stated in the limitations, there was a patient selection bias. In fact, the sample was not randomized.
In addition, it is stated in the study limitations that the study was not originally intended to be carried out online. Therefore, the registered protocol and the ethics committee may not include the information presented here.
CONCLUSIONS
The conclusions are unrealistic and have not taken into account the limitations and biases of the study.
REFERENCES
Many bibliographies are obsolete. The bibliographic citations used are more than 5 years old (49 %). The authors must update and arrange the bibliography.
Author Response
Find them attached in the word.

Reviewer 2 Report
Comments and Suggestions for Authors
The authors propose an interesting aspect of breast cancer treatment management.
I absolutely agree on the importance of physical exercise so this work, at least from this point of view, is very valid.
- I would have used a different tool to evaluate body composition (stratigraphy or plicometry), and I would report the evolution of the phase angle from the bioimpedance which is very important especially in this condition: 10.3390/nu15010050 and 10.23736/S0022-4707.21.12727- 6 for example
- It would be useful to propose training schemes such as suspension training, useful in sarcopenia (10.3389/fspor.2022.950949) but also easy to operate at home
- It would be interesting to know if the subjects practiced physical activity before discovering they had a tumor
Comments on the Quality of English LanguageIt needs revision
Author Response
Find them attached in the word.

Reviewer 3 Report
Comments and Suggestions for Authors
I’ve read with attention the paper of Garcia-Roca et al. that is potentially of interest. The background and aim of the study have been clearly defined. The methodology applied is overall correct, the results are reliable and adequately discussed. The conclusions are consistent with the evidence and arguments presented and they address the main question posed. The references are also appropriate as well as tables and figures. I have no ethical concerns regarding experiments, nor on plagiarism or publication ethics. I’ve only some minor comments:
- The last sentence of the background should be not part of the introduction but of the methods section
- How was the sample size calculated? Is the study sufficiently powered?
- When the p-value is not significant, it should be replaced by the actual p-value
Author Response
Find them attached in the word.

Reviewer 4 Report
Comments and Suggestions for Authors
The article entitled “Effect of a Long-Term Online Home-Based Supervised Exercise Program on Physical Fitness and Adherence of Breast Cancer Patients: A Randomized Clinical Trial” (cancers-2951367) is presented to the section “Clinical Research of Cancer”.
The aim of this study was to analyze the effect of a synchronous-supervised online home-based exercise program (HBG) over 24 weeks on body composition, physical fitness, and adherence compared to an exercise recommendation group (ERG) without supervision, among patients undergoing breast cancer treatment.
The abstract should include the design used, inclusion criteria, and the sample size upon which the study is conducted. Is there a comparison between the groups, or is it a comparison over time? Is the outcome per each control or at the end of follow-up?
In the introduction, it could be noted whether there are studies on both types of exercises proposed in women undergoing breast cancer treatment.
In the materials and methods section, under the study design, it is indicated as a prospective randomized clinical trial with two groups, but it does not detail how patient randomization was carried out in both groups. It should be indicated through a diagram how the follow-up was conducted and whether there were any losses throughout.
In Table 1, when indicating "Luminal A...", it should specify "subtype of breast cancer".
The limitation regarding the part of the study conducted during the pandemic altering the protocol should be addressed. It should be discussed how this situation could affect the results. Age of the patients should also be considered as it could affect the presented results.
The conclusion aligns with the presented results.
Author Response
Find them attached in the word.

Reviewer 5 Report
Comments and Suggestions for Authors
Abstract
This section responds well to what the future reader of this research may be looking for, in which he/she will find a well defined objective, the main characteristics of the sample, what the physical exercise interventions have been, the main results and the general conclusion.
All of this makes potential readers of this article likely to be interested in completing the reading and being quoted in the future.
Introduction
Lines 46-47. If you write "Nowadays"...reference number 1 ...it can't be from 2018! This needs to be checked and changed.
Lines 51-55. You should consider including this quote: Hidrobo Coello, J.F. 2020. Physical activity for patients diagnosed with cancer. Sports prescription guide for Ecuador. Revista Iberoamericana de Ciencias de la Actividad Física y el Deporte. 9, 3 (Dec. 2020), 18-41. DOI:https://doi.org/10.24310/riccafd.2020.v9i3.10100.
Lines 56-57. You should consider including these citations regarding longitudinal exercise treatments in patients who have survived cancer: Gavala-González J, Gálvez-Fernández I, Mercadé-Melé P, Fernández-García JC. Rowing Training in Breast Cancer Survivors: A Longitudinal Study of Physical Fitness. Int J Environ Res Public Health. 2020;17(14):4938. Published 2020 Jul 9. doi:10.3390/ijerph17144938
Gavala-González J, Torres-Pérez A, Fernández-García JC. Impact of Rowing Training on Quality of Life and Physical Activity Levels in Female Breast Cancer Survivors. Int J Environ Res Public Health. 2021;18(13):7188. Published 2021 Jul 5. doi:10.3390/ijerph18137188
The introduction responds well to a recent literature search on the subject and finally defines its objective.
2. Materials and Methods
2.1. Study design
Ok, well explained and with ethical controls included.
2.2. Participants
Ok, very well explained.
2.3. Procedure
Please report the recovery or rest time between each set.
Line 127. Please report exactly what types of aerobic exercises you have used.
Lines 127-128. Please report on how the training programme evolved over the course of the sessions ... and why you did it this way, including supporting literature references.
Lines 1411-142. Please report exactly how you monitored your physical activity levels through phone calls. Do not forget to include the papers that validate this way of monitoring physical activity.
2.4. Outcome assessments
Lines 167-168. Please report how much time you allowed for recovery between each attempt.
2.5. Statistical analysis
Ok, very well explained
3. Results
3.1. Participants characteristics
Table 1. In the variable age you have to put the unit of measurement (years).
3.2. Anthropometry and body composition
Ok, very well explained
3.3. Physical fitness
Ok, very well explained
3.4. Safety and adherence
Ok, very well explained
This is undoubtedly the best section of the article in which the authors make a real effort to compare their results with those of previous research. However, we recommend reviewing the suggested literature in order to further improve this section.
Conclusions
This is not really a conclusion, but a summary explanation of the results. It is hoped that the authors will give some practical recommendations for patients, clinicians and trainers based on the findings of this research, which in a summarised form should be included in this section.
Author Response
Find them attached in the word.

Round 2
Reviewer 4 Report
Comments and Suggestions for Authors
After carefully reviewing the new version of the manuscript entitled “Effect of a Long-Term Online Home-Based Supervised Exercise Program on Physical Fitness and Adherence of Breast Cancer Patients: A Randomized Clinical Trial” (cancers-2951367), as well as the authors' response to the comments and suggestions requested for manuscript improvement, I have confirmed that the authors have satisfactorily addressed the requested information. They provided evidence that a supervised home-based exercise program improved physical fitness compared to an exercise recommendation group in patients undergoing breast cancer treatment. Specifically, the home-based program enhanced performance in the chair stand test, flexibility, squat jump test, countermovement jump test, and 6-minute walking test after 24 weeks of exercise training.